



# 1 The flexural strength of bonded ice

Andrii Murdza[1], Arttu Polojärvi[2], Erland M. Schulson[1], Carl E. Renshaw[1,3]
[1]Thayer School of Engineering, Dartmouth College, Hanover, NH, USA
[2]Aalto University, School of Engineering, Department of Mechanical Engineering, P.O. Box 14100, 00076 Aalto,
Finland
[3]Department of Earth Sciences, Dartmouth College, Hanover, NH, USA
*Correspondence to*: Andrii Murdza (Andrii.Murdza@dartmouth.edu)
**Abstract.** The flexural strength of ice surfaces bonded by freezing, termed freeze-bond, was studied by performing
four-point-bending tests of bonded freshwater S2 columnar-grained ice samples in the laboratory. The samples were
prepared by milling the surfaces of two ice pieces, wetting two of the surfaces with water of varying salinity, bringing
these surfaces together, and then letting them freeze under a compressive stress of about 4 kPa. The salinity of the
water used for wetting the surfaces to generate the bond varied from 0 to 35 ppt. Freezing occurred in air under
temperatures varying from -25 to -3 °C over periods that varied from 0.5 h to ~100 hours. Results show that an increase
in bond salinity or temperature leads to a decrease in bond strength. The trend for the bond strength as a function of
salinity is similar to that presented in Timco and O'Brien (1994) for saline ice. No freezing occurs at -3 °C once the
salinity of the water used to generate the bond exceeds ~25 ppt. The strength of the saline ice bonds levels off (i.e.,
saturates) within 6-12 hours of freezing; bonds formed from fresh water reach strengths that are comparable or higher
than that of the parent material in less than 0.5 hours.
**1. Introduction**

20       Freeze bonds form when distinct ice features, such as floating ice floes or ice blocks of a rubble pile, become

and remain in contact over a period of time at low enough temperature. Insight into the strength of the bonds is
important when, for example, the strength of an ice cover formed of refrozen floes or the strength of an ice rubble pile
is estimated. There are several factors that affect the failure of a cover of sea ice, surface waves being a major one
that has gained an increasing amount of interest recently (Shen, 2017; Squire, 2020). Under the action of waves, ice
covers bend and may undergo flexural failure (Ardhuin et al., 2020; Asplin et al., 2012; Collins et al., 2015; Hwang
et al., 2017; Kohout et al., 2014, 2016; Shackleton, 1982). It is relevant to ask if the freeze bonds within a broken and
refrozen ice cover form the weakest link at which wave-induced cracks initiate and propagate. During the wave-ice
interaction, the freeze bonds deform and failure occurs under a tensile state of stress arising from flexural deformation.
To our knowledge, no data on freeze-bond strength under tensile loading have been published.

The strength of freeze bonds has been tested only under combined compressive and shear loading. Such tests

have been related to continuum modeling of ice rubble using material models having yield surfaces resembling that





of a Mohr-Coulomb material model (Ettema and Urroz, 1989; Heinonen, 2004; Liferov et al., 2002, 2003; Serré,
2011b, 2011a). The critical shear stress of a Mohr-Coulomb material is given by $\tau = c + \sigma \tan \varphi$, where $c$ is the
cohesion, $\sigma$ the compressive stress, and $\varphi$ the internal friction angle of the material. The underlying assumption in
testing has been that the failure of the individual freeze bonds within the rubble occurs through the same mode as the
failure of the rubble itself. No evidence of this type of similarity between the two scales exists. Instead, the numerical
simulations (Polojärvi and Tuhkuri, 2013) suggest that the individual freeze bonds within deforming rubble do not
fail due to shear, but rather under tensile stresses as the bonded ice blocks move relative to each other. This implies
that data on the shear strength of the freeze bonds may not lead to reliable estimates of the shear strength of ice rubble.

In this paper, the strength of freeze bonds under tensile loading is studied. For this purpose, we conducted
four-point-bending tests using the apparatus described and used by Murzda et al. (2020). All procedures for testing
were designed with the aim of reducing the number of variables for reliable analysis: bonds were formed between
milled surfaces of freshwater ice specimens (termed the parent material) and bond freezing and testing were performed
in air under a small compressive stress of about 4 kPa. The experimental variables were the freezing time (0.5 h…~100
h), the sample temperature (-3°C…-25°C), and the salinity of the water used to form the bond (0…35 ppt). Bond
strength initially increases with freezing time, but then appears to level off and to reach a plateau (i.e., to saturate)
over several hours. Depending on the salinity of the water from which the bond is formed, the saturation time for bond
strength ranges from 0.5 h to 12 h. The "saturated strength" of freshwater bonds with finer microstructure appears to
reach levels higher than the strength of the parent material with a larger grain size. The results from these experiments,
presented below, represent the first set of results on the failure of freeze bonds under tension.
**2. Experimental procedure**
Freshwater ice, used here as the parent material for the freeze-bonded samples, was produced in the
laboratory as described in Smith and Schulson (1993) and Golding and others (2010). Tap-water was frozen
unidirectionally from top to bottom in a cylindrical 800 L polycarbonate tank, forming pucks of ~1 m in diameter.
The ice was generally bubble-free and columnar-grained. Thin-section analysis showed that the average column
diameter, as measured in the horizontal plane normal to the direction of ice growth using the linear intercept method,
was 5.5±1.3 mm. The c-axes were randomly oriented within, and confined to, the horizontal plane, suggesting that the
ice had an S2 growth texture (in the terminology of Michel and Ramseier, 1971). The ice density was 914.1±1.6 kg·m⁻
³ (Golding and others, 2010); Young's modulus in the horizontal plane was 9.52 GPa (Snyder and others, 2016). Once
grown, the ice was cut into blocks and stored in plastic cooler boxes in a cold room at -10° C. Specimen preparation
is described in detail elsewhere (Iliescu et al., 2017; Murzda et al., 2018, 2019, 2020, n.d.).

Samples to be freeze-bonded were prepared from the ice blocks by milling them into thin plates. The plates
had dimensions of $h$ ~15 mm in thickness (parallel to the long axis of the grains), $b$ ~ 85 mm in width, and $l$ ~300 mm
in length. Specimens were allowed to equilibrate to the test temperature for at least 24 hours prior to testing.



| | |
|---|---|
| 68 | |
| 69 | The plates were then cut perpendicular to their long axis into two parts. In most samples the sawn surfaces |
| 70 | were milled after cutting (more below). To initiate freeze-bond growth, the sawn and milled surfaces were sprayed |
| 71 | with a fine mist of water at a temperature of +2°C and quickly brought into contact by setting the two pieces into a |
| 72 | freeze-bonding rig (Figure 1) in a cold room. The surfaces were wet when brought into contact, but in addition, a |
| 73 | syringe was used to inject about 0.1 ml of water to the bond to ensure uniform wetting of the surfaces. Excess water, |
| 74 | if any was observed around the bond, was wiped with a tissue. All of the above steps were performed at +2 °C to |
| 75 | prevent freezing from occurring before setting the sample into the rig. |
| 76 | |
| 77 | To investigate whether the roughness of the faces in contact affects the bond strength, a few samples had |
| 78 | their faces produced by cutting the parent plate either with a coarse (1/2 inch in width, 1/40 inch in thickness and |
| 79 | 6 teeth per inch) or a fine (13/64 inch in width, 1/64 inch in thickness and 24 teeth per inch) band saw. Although few |
| 80 | in number, results from these initial experiments suggested that surface roughness of the kind we explored had no |
| 81 | significant effect on flexural strength. Thus, for all further tests (that led to the results reported below) sawn surfaces |
| 82 | were milled for consistency and reproducibility (more in Discussion). |
| 83 | |
| 84 | Figure 1 shows a sketch (a) and photograph (b) of the freeze-bonding rig. The rig had a system consisting of |
| 85 | two plastic bars and two springs for applying a desired pressure (i.e., compressive stress) to the bond during freezing. |
| 86 | In the present experiments, a confining pressure of ~4 kPa was chosen which is in accordance with the maximum |
| 87 | hydrostatic pressure within submerged 10-meter-thick ice rubble mass (Ettema and Schaefer, 1986). The rig was kept |
| 88 | in a cold room of the desired temperature (i.e. from -25°C to -3°C) during freezing. The base of the rig was made from |
| 89 | an acrylic plate having low heat conductivity, ensuring the heat flux in the bond area was mainly along the long axis |
| 90 | of the sample. Wax paper was placed between the ice and the acrylic to prevent freezing of ice onto the rig. All |
| 91 | materials of the rig were such that the frictional resistance between them and ice was low. This enabled good control |
| 92 | of the confining pressure and sample alignment. |
| 93 | |
| 94 | To investigate the effect of the salinity on the bond strength, fresh water and saline water of salinity ranging |
| 95 | from 2 to 35 ppt (parts per thousand), was used in spraying. Saline water was prepared in the manner described by |
| 96 | Golding et al. (2010, 2014) by adding the commercially available salt mixture "Instant Ocean" to tap water. Salinity |
| 97 | was measured using a calibrated YSI Pro30 conductivity salinity meter. |
| 98 | |
| 99 | After a desired time of freezing, varying from 0.5 to ~100 h, the freeze-bonded sample was removed from |
| 100 | the rig and its flexural strength under four-point bending was measured. For this purpose, a servo-hydraulic loading |
| 101 | system (MTS model 810.14) with a custom-built four-point loading frame was utilized. The sketch of the apparatus |
| 102 | is shown in Figure 2 of Murdza et al. (2020), the photograph of the apparatus is shown in Figure 5a and the apparatus |
| 103 | is described in detail elsewhere (Iliescu et al., 2017; Murdza et al., 2018, 2019, 2020). The outer loading rollers are |
| 104 | immobile during testing while the inner loading rollers are attached to the actuator. The hydraulic actuator was driven |





under displacement control and loading was controlled using a FlexTest-40 controller. A calibrated load cell was used
to measure the load.

The experiments were performed at an outer-fiber center-point displacement rate of 0.1 mm s$^{-1}$ (or outer-

fiber strain rate of about 1.4 x 10$^{-4}$ s$^{-1}$). This displacement rate resulted in an outer-fiber stress rate of about 1 MPa s$^{-}$
$^{1}$. As was indicated earlier (Murdza and others, 2020), the 0.1 mm s$^{-1}$ displacement rate in cycling results in a period
of ~20 s which is approximately the frequency of ocean swells (Collins and others, 2015).  The major outer-fiber stress
$\sigma_f$ was calculated as:

$$\sigma_f = \frac{3PL}{4bh^2} \ , \hspace{5cm} (1)$$

where $P$ is the applied load and $L$ is the distance between the outer pair of loading cylinders and is set by the geometry
of the apparatus to be $L$ = 254 mm. It is important to note that in all experiments described in this paper the bond
formation and breaking of bonded ice occurred at the same temperature. Owing to the confining impact of the loading
cylinders of the 4-point flexing apparatus (see Figure 5a and Figure 2 of Murdza et al. (2020)) and to the Poisson
effect, a biaxial state of tension developed in the ice. Based on isotropic elasticity and plasticity theories, the minor
stress was approximately between one-third to one-half of the major stress.

## 3. Results and Observations

### 3.1. Flexural strength of parent material

Two measurements on the flexural strength of pristine ice plates, that is, plate-like samples of parent material

without a freeze bond, were conducted at -10 °C. The strength values obtained were 1.51 and 1.63 MPa. Only two
experiments were performed as these values compare favorably with the earlier measurements by Murdza et al. (2020)
on the same kind of ice using the same loading system. Murdza et al. (2020) reported that the average and the standard
deviation of the flexural strength at -3, -10 and -25 °C were 1.42±0.16, 1.67±0.22 and 1.89±0.01 MPa, respectively.
Further, the measured values are in agreement with the data that are reviewed in Timco and O'Brien (1994), where
the average and standard deviation of 1.73±0.25 MPa is reported for the flexural strength of freshwater ice at
temperatures below -4.5 °C.

### 3.2. Flexural strength of bonded ice

### 3.2.1. Freshwater bond

The experiments with a freshwater bond were conducted at -3 and -10 °C. The results are listed in Table 1.

The time for the bond formation (0.5 hours was the shortest freezing period used here, implying that the bond formed
in less time) is reasonably consistent with analytical estimates, Appendix A. Surprisingly, in all of these experiments,
the failure occurred outside of the bond. This suggests that even after only a relatively short period of freezing, the
strength of the freshwater bond reaches and exceeds that of the parent material. Even though the results listed in Table



1 show scatter, at -10°C comparison of the measured flexural strengths to those described in Section 3.1 showed that
they are not statistically different from the flexural strength of pristine freshwater ice samples (*p-value* = 0.21 and 0.08
for tests at -3 and -10 °C, respectively). This is important because it indicates that the above-described bond generation
procedure did not hamper the samples by, for example, leading to geometrical misalignments in them.
**3.2.2. Saline bond**

Figures 2 and 3 show the results from the experiments performed to investigate the effect on bond strength

of the salinity of the water used to create the freeze bond. The data are given in Tables 2-4. The tables indicate the
experiments where no freezing occurred ("No") and the experiments where bonding occurred, but the bond was too
weak to be tested ("Low"). These data are excluded in the figures below.

Figure 2 shows that the strength of the saline bonds increases over time and levels off, or saturates, after

about 6-12 h. Thus, the strength of the saline bonds increases at a considerably lower rate than that of the freshwater
bonds. The reason is likely related to the rejected salts and entrapped air at the ice-water inferface that slows the rate
of the interface advance. A comparison of these results to those in Section 3.1 shows that the strength of the saline
bonds is well below the strength of the freshwater ice used as the parent material.  A comparison of the two data sets
in Figure 2 shows that the saturated strength of the bonds made from water of higher salinity but at lower  temperature
(35 ppt and -10 °C) is about twice of that of the bonds with lower salinity but higher temperature (12 ppt and -3 °C).

Figure 3 illustrates how the salinity of the water used to generate the freeze-bond at -3 °C affects its saturated

strength at -3 °C. While the measured strength values for low salinities are close to those measured for freshwater ice,
the bond strength decreases rapidly with an increase in salinity and no freezing occurs once the salinity of the salt
water used to generate the bond reaches ~25 ppt; even at 17 ppt some bonds were too weak to be tested. This agrees
reasonably well with analytical estimates, Appendix B, where formulas that relate strength to volume fraction of solid
phase suggest that at salinities of 30 ppt and above at -3 °C no freezing occurs. Figure 3 additionally shows two
exponential fits, one directly fitted to our data and one by Timco and O'Brien (1994) for the flexural strength of saline
ice (equations for these fits provided in Appendix C, where $\sigma_b$ is flexural strength in MPa and $\nu_b$ is liquid brine content
in parts per thousand). It is important to notice that the fit by Timco and O'Brien (1994) yields lower values than the
measured bond strength in the present study for the whole range of salinities used. Likewise, the actual strength values
for the freshwater bonds are greater than the ones suggested by Figure 3, since the failure in these cases occurred
outside the bond, indicating that the bond is stronger than the parent material. Both saline and freshwater bonds that
develop through freezing appear to reach strengths higher than that of S2 type parent material of the same salinity
(strength of saline parent material is assumed to be the same as in Timco and O'Brien (1994)).

Temperature has a strong effect on the saturated strength of the freeze bonds. Figure 4 and Table 5 summarize

the data from experiments on specimens having bonds made from water of salinity 20 ppt at temperatures  from -3 °C
to -25 °C. Three out of the four specimens at -25 °C failed outside of the bond with a measured strength of
1.61±0.12 MPa, which is close to 1.89 MPa measured earlier at -25 °C on the same type of freshwater ice (Murdza
and others, 2020). Figure 4 also suggests that no freezing occurs at temperatures above about -3 °C, which is in fair





agreement with analytical estimates of no strength at $T = -2$ °C in Appendix B. Though the analytical equation from
Appendix B predicts well when no freezing occurs, it does not yield a trend that describes most of the data in Figure
4. The reason may be that for the microstructure of bonds in the present study, strength may not be directly proportional
to volume fraction of the solid phase as the model   in Appendix B assumes, but rather a non-linear function of the
volume fraction of solid.
Figure 5a-c show an example of the typical samples after failure. Figure 5a shows a case where the crack had
initiated at the bond and started to propagate along it, but then deviated from it and continued to grow through the
parent material. Figure 5b shows a close up of a bond face-on after the most common type of failure, which occurred
along the bond. In this case, both surfaces of the failed freeze-bond had a fairly uniform "blurry" appearance which
indicates that failure occurred through the ice of the bond. It was also fairly usual for the samples having low salinities,
low temperatures and long freezing times, that the crack initiated and started to propagate along the bond and then
slightly deviated and moved parallel to the bond but inside the parent material, as shown by Figure 5c.
**4. Discussion**
The above results are the first measurements to be reported for the strength of freeze bonds under tensile
loading. Although the experiments were performed under flexural loading, they provide unique data on the tensile
strength of the freeze bonds. Under the loading conditions, the flexural strength of ice is governed by tensile strength,
although measured strengths are greater by a factor of about 1.7 than strengths measured under pure tensile loading
(Ashby and Jones, 2013). Murdza and others (2020) showed that the flexural strength of freshwater S2 ice tested on
the same loading system as used here compares well with direct measurements of the tensile strength of the same type
of ice at the same conditions (Carter, 1971) when divided by 1.7. By using this factor to scale the values for saturated
bond strengths shown in Figure 2 leads to tensile strength values of about 0.3 MPa and 0.18 MPa for bonds at -10 °C
and -3 °C, respectively.

While there are no other data on the tensile strength of freeze bonds, the results can be compared to the
relatively large amount of earlier work on the shear strength of  freeze bonds (Bailey et al., 2012; Boroojerdi et al.,
2020a, 2020b; Bueide and Høyland, 2015; Ettema and Schaefer, 1986; Helgøy et al., 2013a, 2013b; Marchenko and
Chenot, 2009; Repetto-Llamazares et al., 2011b, 2011a; Shafrova and Høyland, 2008). Common values for the shear
strength in those studies ranged from 0.01 to 0.1 MPa, which are considerably lower than the flexural strength values
measured here. Usually, these strengths have been measured for bonds grown under water over periods that have not
been long enough to reach saturated bond strengths. On the other hand, the highest reported shear strength values
~0.3…0.7 MPa (Bailey et al., 2012; Boroojerdi et al., 2020b; Shafrova and Høyland, 2008) are within the same range
as the flexural strength values measured here. Given that the tensile strength of ice is, on average, lower than the shear
strength (Timco and Weeks, 2010), the strength values measured here are perhaps surprisingly high. The high strength
values here likely relate to the well-controlled bond growing procedure and possibly to a finer microstructure of the
material that comprises the bond.




208   Work on the shear strength of freeze bonds has led to a conclusion that the evolution of the bond strength has

209 three phases (Boroojerdi et al., 2020b; Repetto-Llamazares et al., 2011b, 2011a): (1) an initial period of a few minutes

210 of increasing strength due to heat flux from the bond to the parent material, (2) a period of some hours of weakening

211 as the temperature of the bond increases due to water surrounding it and (3) a period of several days of strengthening

212 due to sintering. The evolution of the flexural strength of the bonds in the present experiments is likely similar to that

213 of phases (1) and (3). The initial bond strengthening can be related to the transfer of heat along the long axis of the

214 specimen and the accompanying advance of the ice/water interface. Given that the water layer after wetting the contact

215 surfaces is very thin, the bond strength would be expected to saturate quickly; Appendix A describes a simple model

216 and suggests that the process, similar to above described phase (1), takes fewer than 10 minutes at -10 °C and a greater

217 amount of time of about 1.5 h at -3 °C, aligning with earlier studies and the result here. This means that for saline

218 bonds, phase (3) has a duration of about 6…12 hours, whereas earlier experiments have occasionally had relatively

219 long freezing times, varying from 60 h to 12 days (Bailey et al., 2012; Shafrova and Høyland, 2008).

220

221   As part of the studies on the evolution of bond strength, it has been fairly common to investigate the ratio of

222 the bond strength values to that of the parent material (Bailey et al., 2012; Boroojerdi et al., 2020b; Shafrova and

223 Høyland, 2008). Shafrova and Høyland (2008) found that specimens with bonds grown in the field had the strength

224 ratio varying from 0.008 to 0.082 (with a mean of 0.03 after 48 hours of bonding). For laboratory-grown bonds, they

225 measured ratios in the range 0.06 to 0.69 ($0.21 \pm 0.12$). The latter values are in line with values reported by Bailey et.

226 al. (2011) and Boroojerdi et al. (2020b), who reported ratios up to about 0.70 and 0.85, respectively. Boroojerdi et al.

227 (2020b) suggested an empirical formula to describe the strengthening of a freeze bond during the above-described

228 phase (3). The formula was based on curve fitting and an assumption that the shear strength of the bond approaches

229 asymptotically that of the parent material with increasing sintering time. The experiments here indicate that such an

230 assumption may not be always justified, as at least the flexural strength of the freeze bonds can reach values that are

231 above that of the parent material.

232

233   Ettema and Schaefer (1986) and Repetto-Llamazares et al. (2011b) studied whether freezing in the presence

234 of water has an effect on the shear strength of the freeze-bond. The results indicate that shear strength was higher

235 when bonds froze under water. While Ettema and Schaefer (1986) let the bonding occur with samples submerged in

236 fresh water, Repetto-Llamazares et al. (2011b) used 7 ppt saline water for submerging. Earlier studies on the effect of

237 freezing conditions have not had the opposing surfaces wetted before bringing them together when generating bonds

238 in air. This effectively removes the above-described phase (1) from the bond strength evolution, if the result of the

239 heat transfer during the initial period of bond strengthening is assumed to simply be freezing of the liquid at the bond

240 interface. In addition, in these earlier studies, the maximum freezing times for the bonds grown in air varied only

241 from 0.5 min to 3 min. As phase (3) takes at least several hours, it seems likely that the mentioned studies have not

242 yielded data on saturated bond strengths for bonds grown in air.






While the new data from the present tests yielded clear trends for the strength of the freeze-bonds, they also
showed significant scatter. This scatter, even when bond generation was performed in a simplified manner using
carefully prepared milled samples (Section 2), is an indication that the strength of freeze bonds is a parameter that
inherently shows wide scatter. One reason for this, amongst others perhaps, is the detailed microstructure/phase
distribution of the bond. The microstructure probably varies somewhat from specimen to specimen, thereby leading
to variations in bond strength. The variation is actually of similar magnitude to that observed in experiments on the
flexural strength of pristine ice samples made with the same apparatus (Murdza et al., 2020).
The fact that in samples with freshwater bonds failure initiated and propagated outside of the bond suggests
that the strength of the freshwater bond is greater than the strength of pristine freshwater S2 ice. This may indicate a
difference in microstructure between the ice in the freeze bond and the ice of the parent plates. A finer grain size
within the bonds may be due to the initial water layer, which was produced by spraying a very fine mist, creating small
water droplets working as nucleation sites for the ice grains in the bond. Our argument is supported by the work of
Schulson and others (1984) who showed that tensile strength strongly depends on the grain size, increasing as grain
size decreases. A difference in grain size could also explain the fact that the strength versus salinity curve from Timco
and O'Brien (1994) is below the trend obtained in the present study (Figure 3). Concerning of the microstructure of
the bonds, one may think that because one phase is dominant it should form the matrix; however, there is at least one
class of materials, namely high-temperature nickel-based superalloys (Sims, 1984) where the minor component forms
the matrix. Since we do not know the structure of the bonds in the present study, we cannot be conclusive in this
regard.
As one may have expected, both temperature and salinity affect the flexural strength of bonded ice samples.
The trend of strength versus salinity (Figure 3) has an exponential functionality similar to what has been suggested by
Timco and O'Brien (1994), while the trend of strength versus temperature (Figure 4) appears, to a first approximation,
to be roughly linear. It is important to mention here that the salinities provided in this paper are salinities of the spray
and not of melt-water from the bond itself, and this begs the question: Is the bond salinity the same or lower than the
salinity of spray solution? In the formation of a natural floating sea ice cover, of course, the rejection of salts from
ice results in melt-water salinities lower than bulk water salinity. (Weeks and Ackley, 1986) . Given that in our
experiments the bond thickness is very small (<1 mm) and freezing time is relatively short, it is unlikely that all the
salt is expelled from the freeze bond, resulting in the bond salinity similar to the spray salinity. Therefore, while the
resulting bonds might have salinities slightly lower than the sprayed water, the results yield a reasonably reliable trend
of strength as a function of salinity which is very similar to the relationship proposed by Timco and O'Brien (1994).
The effect of surface roughness at the freezing interface was also briefly addressed when performing the
experiments. In addition to the milled surfaces with a roughness of $0.43 \pm 0.24 \times 10^{-6}$ m in the direction of milling and
$2.01 \pm 0.47 \times 10^{-6}$ m in the orthogonal direction (Schulson and Fortt, 2012), experiments were performed on samples
with surfaces produced by using a fine and coarse band saw blade, which resulted in ice surface roughness of up to



~1 mm. The results from the experiments with differently produced surfaces showed no significant difference on the
strength of freshwater and saline bonds (1.39 MPa vs 1.43±0.15 MPa for freshwater bonds and 0.39±0.13 MPa vs
0.34±0.16 MPa for saline bonds of 12 ppt salinity). As milling could be performed with the highest accuracy from the
aspect of sample dimensions and alignment, it was chosen as the technique we used here. Unlike what was observed
in the present study, Helgøy et al. (2013a) observed that the surface roughness does affect freeze bond shear strengths,
with rougher surface leading to bonds having higher strength. The discrepancy between their results and results in the
present study suggests that there may exist a threshold value for the surface roughness, after which it affects the bond
strength; it is likely that both milled and sawn surfaces used in the present study are too smooth for the effect of surface
roughness to be observed. On the other hand, experiments on shear strength usually involve sliding motion between
the blocks of the parent material. This motion may become restricted by rough surfaces, which could lead to higher
shear loads interpreted to be due to an increase in freeze-bond strength. In tests under tensile loading, such kinematic
restrictions do not exist.
Finally, it is worth noting that while this is the first study on the flexural strength of freeze bonds, it is not
the complete story. Further work is needed to investigate the effects of other factors such as bond pressure, the
character of parent ice plate, bond microstructure, the width of the opening to be bonded, etc.
**5. Conclusions**
Systematic experiments on the flexural strength of freeze bonds were conducted for the first time. The bonds
were grown in the air under 4 kPa confining pressure. The parent material was S2 columnar-grained freshwater ice.
The salinity of the bond varied from 0 to 35 ppt and freezing temperatures from -3 to -25 °C. It is concluded that:
(i) Freshwater bond strength exceeds the strength of parent ice in less than 0.5 h upon freezing.
(ii) The saline bonds reach their saturated strength within about 6-12 h of freezing.
(iii) An increase in bond salinity and in freezing temperature leads to a decrease in bond strength.
(iv) The relationship between bond strength and its salinity is similar to the one suggested by Timco and O'Brien
305 (1994).
(v) No freezing occurs once the salinity of the water used to generate the bond reaches values of about ~25 ppt
at -3 °C.
**Acknowledgements**
The authors are grateful for the financial support from the Academy of Finland through the project no. 309830 ("Ice
Block Breakage: Experiments and Simulations (ICEBES)") and National Science Foundation (FAIN 1947-107). Arttu
Polojärvi worked on the article while visiting Thayer School of Engineering at Dartmouth College (Hanover, NH,





USA) during spring 2020, thanks are extended to Prof. Erland Schulson for hosting. Finnish Maritime Foundation is
acknowledged for partial funding of the visit.
**Author contributions:** AM, AP, ES, and CR designed the experiments, and AM carried them out. AM and AP
prepared the manuscript with contributions from all co-authors.
**Competing interests:** The authors declare that they have no conflict of interest.

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

**Appendix A: Time for the freeze-bond formation**
To estimate the time to form a freeze-bond we assume that heat fluxes are along the long axis of the sample,
i.e. horizontal temperature gradients are much larger than vertical gradients at the freezing interface. The other
assumptions are that the heat flow through a material is equal to the energy for the solidification of the water along
the bond and that no heat losses occur, i.e.

$$k\frac{dT}{dx} = \rho\lambda\frac{dx}{dt} \qquad (A1)$$


where $t$ is the time, $T$ is the ice temperature, $k$ is the thermal conductivity of ice, $\lambda$ is the latent heat of fusion of ice
per unit mass, $\rho$ is the ice density and $\Delta x$ is the characteristic conduction length or, in our case, the thickness of the
bond.
Taking into account that thermal diffusivity $\alpha = {k}/{\rho c_p}$, where $c_p$ is specific heat capacity, and that $x = \sqrt{\alpha t}$,
after integration of Equation A1 we obtain the relationship:

$$t = \frac{1}{4}\left(\frac{x\lambda}{\Delta T}\right)^2\left(\frac{\rho}{kc_p}\right) \qquad (A2)$$


A note of caution is necessary here. As ice-water interface advances during freezing in saline ice, both air
and salt are rejected and build up at the inferface. Unlike freezing in nature, there is not enough space for rejection
and, as a result, this slows the rate of advance of the interface.

According to Equation A2, and using parametric values of $c_p = 2100\,J/kg°C$, $\lambda = 330\,kJ/kg$, $k =$
$2.2\,W/m°C$, $\rho = 914\,kg/m^3$ for freshwater ice at -10 °C and bond thickness of 1 mm we need only 1 min for the
bond formation, while for freshwater ice at -3 °C a similar bond forms in about 10 minutes. While this estimate is
consistent with observations, it is also in accord with earlier experimental results by Repetto-Llamazares et al. (2011a,
2011b) and Borojeerdi et al. (2020a) for phase (1) increase during freeze bond shear strength evolution (Section 4).

**Appendix B: The strength of freeze-bonds as a function of salinity and temperature**
Principle:

The freeze bond is comprised of essentially two phases, solid (ice) plus liquid (water), barring entrapped air.
To a first approximation, we assume that its strength, $\sigma_{fb}$, is proportional to the volume fraction, $f_s$, of the solid phase.
The constant of proportionality, $\sigma_{f0}$, is the strength of freshwater ice. The relationship:





$$\sigma_{fb} = \sigma_{f0} f_s . \tag{A3}$$

The volume fraction of the solid phase is obtained from the lever rule:

$$f_s = \frac{X_l - X_0}{X_l - X_s} , \tag{A4}$$

where $X_l$ and $X_s$ denote the limit of solubility of salt in the liquid (water) and in the solid (ice) phases, respectively,
and $X_0$ is the concentration of salt in the water before freezing is initiated. Over the temperature range of interest, the
phase diagram for the $H_2O$-NaCl system (i.e., thermodynamics) dictates that both $X_l$ and $X_s$ increases with decreasing
temperature, T, according to the relationships:

$$X_l = \frac{T - T_0}{m_l} , \tag{A5}$$

$$X_s = \frac{T - T_0}{m_s} , \tag{A6}$$

where $T_0$ denotes the melting point of "pure" ice (273 K) and $m_l$ and $m_s$, respectively, denote the slope of the liquidus
and the solidus on the phase diagram; both slopes are negative. The solubility of salt in ice is very low and so for
practical purposes $X_s \sim 0$. Writing the temperature difference as $T - T_0 = \Delta T$, the volume fraction of ice within the
freeze bond from Eqn (A4) is given by:

$$f_s = \left(1 - \frac{m_l X_0}{\Delta T}\right) . \tag{A7}$$

Thus, upon equating $X_0$ to salinity, $S$, the strength of the freeze bond is given by:

$$\sigma_{fb} = \sigma_{f0} \left(1 - \frac{m_l S}{\Delta T}\right) . \tag{A8}$$

Taking $m_l$ to be independent of concentration, its value is $m_l = -0.1 \, Kpsu^{-1}$ , giving:

$$\sigma_{fb} = \sigma_{f0} \left(1 + \frac{0.1 S}{\Delta T}\right) , \tag{A9}$$

where $\Delta T < 0$.

The model thus dictates that once freezing is complete the strength of the freeze bond decreases linearly with
increasing salinity, reaching the limit of zero strength when $S = \Delta T / -m_l$.

Both dictates are in reasonable agreement with observation.

**Appendix C: Trends in Figure 3**

The red trend in Figure 3 is taken from (Timco and O'Brien, 1994) where the authors report values for
flexural strength of saline ice over the range of salinities used in the present study and for temperatures above -4.5°C
($\sigma_f$ in MPa), i.e.




$$\sigma_f = 1.76 e^{-5.88\sqrt{v_b}}. \tag{A10}$$



To calculate salinity $S$ (in ppt) based on the liquid brine content $v_b$ (brine volume fraction) in Timco and
O'Brien (1994) we used the following relationship suggested by (Frankenstein and Garner, 1967):

$$v_b = S\left(\frac{49.185}{|T|} + 0.532\right) \tag{A11}$$


where $T$ is the ice temperature in degrees Celsius between -0.5 °C and -22.9 °C. The fit to our data in Figure 3 (black
curve) was made according to the least square method which resulted in the following equation ($\sigma_f$ in MPa):

$$\sigma_f = 1.12 e^{-5.88 v_b} \tag{A12}$$




























**Table 1. Results from testing freshwater bond experiments. The time here is the bond formation time, the strength is the**
**flexural strength (temperature during flexural testing and bond formation was the same). The reader should notice that in**
**all of these experiments the failure occurred outside of the bond and within the parent material.**

| Sample # | Temperature [°C] | Time [h] | Strength [MPa] |
|---|---|---|---|
| 1 | -10 | 24 | 1.43 |
| 2 | -10 | 25 | 1.39 |
| 3 | -10 | 24 | 1.28 |
| 4 | -10 | 3 | 1.58 |
| 5 | -3 | 1.5 | 1.02 |
| 6 | -3 | 1.5 | 1.28 |
| 7 | -3 | 0.5 | 1.4 |


**Table 2. Results from testing saline bond experiments at -10 °C and 35 ppt.**

| Sample # | Time [h] | Strength [MPa] |
|---|---|---|
| 8 | 1.5 | 0.15 |
| 9 | 3 | 0.1 |
| 10 | 26 | 0.34 |
| 11 | 34 | 0.54 |
| 12 | 25 | 0.64 |
| 13 | 82 | 0.61 |
| 14 | 6 | 0.38 |
| 15 | 12 | 0.54 |


**Table 3. Results from testing saline bond experiments at -3 °C and 12 ppt.**

| Sample # | Time [h] | Strength [MPa] |
|---|---|---|
| 16 | 1.5 | Low |
| 17 | 1.5 | 0.31 |
| 18 | 3 | 0.17 |
| 19 | 3 | 0.18 |
| 20 | 6 | 0.25 |
| 21 | 6 | 0.22 |
| 22 | 14 | 0.48 |
| 23 | 24 | 0.14 |
| 24 | 24 | 0.35 |
| 25 | 72 | 0.29 |
| 26 | 97 | 0.32 |







**Table 4. Results from testing saline bond experiments at -3 °C.**

| Sample # | Salinity [ppt] | Time [h] | Strength [MPa] |
|----------|----------------|----------|----------------|
| 27 | 35 | 1.5 | No* |
| 28 | 35 | 24 | No* |
| 29 | 25 | 24 | No* |
| 30 | 20 | 28 | 0.12 |
| 31 | 17 | 3 | Low* |
| 32 | 17 | 13 | Low* |
| 33 | 17 | 25 | 0.3 |
| 34 | 17 | 113 | 0.28 |
| 35 | 10 | 24 | 0.34 |
| 36 | 10 | 24 | 0.34 |
| 37 | 10 | 24 | 0.41 |
| 38 | 10 | 26 | 0.77 |
| 39 | 10 | 73 | 0.54 |
| 40 | 5 | 21 | 0.37 |
| 41 | 5 | 24 | 0.46 |
| 42 | 5 | 24 | 0.75 |
| 43 | 2 | 25 | 0.62 |
| 44 | 2 | 24 | 0.91 |

* "No" and "Low" correspond to "No freezing occurred" and "Strength was too small to be measured", respectively.

**Table 5. Results from testing of ice with bond salinity of 20 ppt after ~24 h of freezing.**

| Sample # | Temperature [°C] | Strength [MPa] |
|----------|------------------|----------------|
| 45 | -25 | 1.69* |
| 46 | -25 | 1.67* |
| 47 | -25 | 1.47* |
| 48 | -25 | 1.13 |
| 49 | -20 | 1.25 |
| 50 | -20 | 0.71 |
| 51 | -15 | 0.87 |
| 52 | -15 | 0.76 |
| 53 | -15 | 0.63 |
| 54 | -15 | 0.55 |
| 55 | -15 | 0.35 |
| 56 | -15 | 0.2 |
| 57 | -10 | 0.66 |
| 58 | -10 | 0.64 |





| Sample # | Temperature [°C] | Strength [MPa] |
|----------|------------------|----------------|
| 59 | -10 | 0.46 |
| 60 | -10 | 0.4 |
| 61 | -5 | 0.2 |
| 62 | -5 | 0.1 |
| 63 | -3 | 0.12 |

*failure occurred outside the bond.

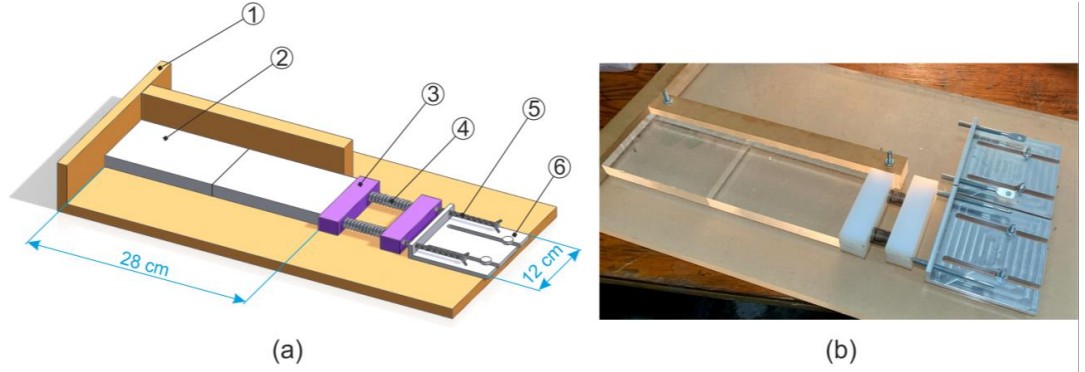

**Figure 1. Sketch (a) and photograph (b) of the freeze-bonding rig with an ice sample having the shape of a thin plate: 1 –**

**acrylic plate; 2 – ice specimen; 3 – plastic bar; 4 – spring; 5 – bolt; 6 – fixation plate.**



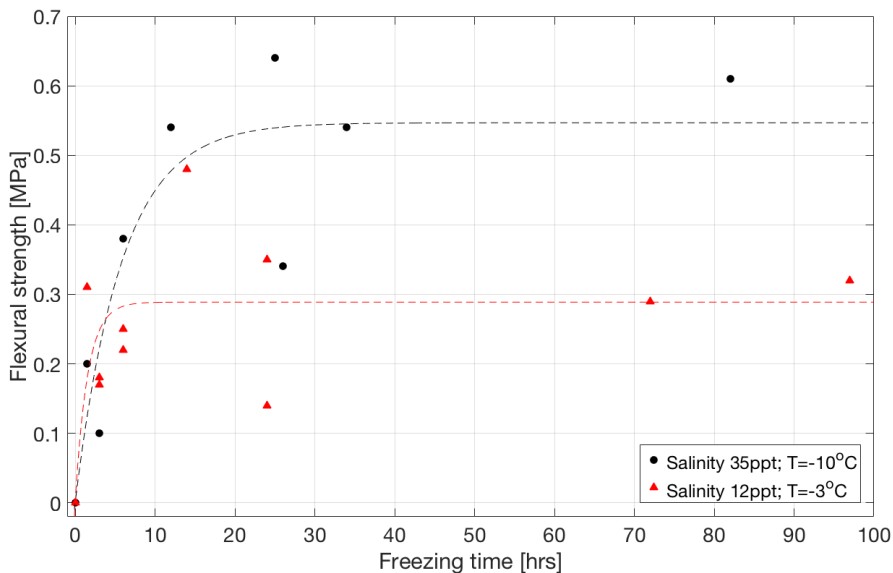


**Figure 2. Flexural strength as a function of freezing time for bonded ice prepared from salt water of 35 ppt salinity at -10°C**

**(in black) and from salt water of 12 ppt salinity at -3°C (in red).**

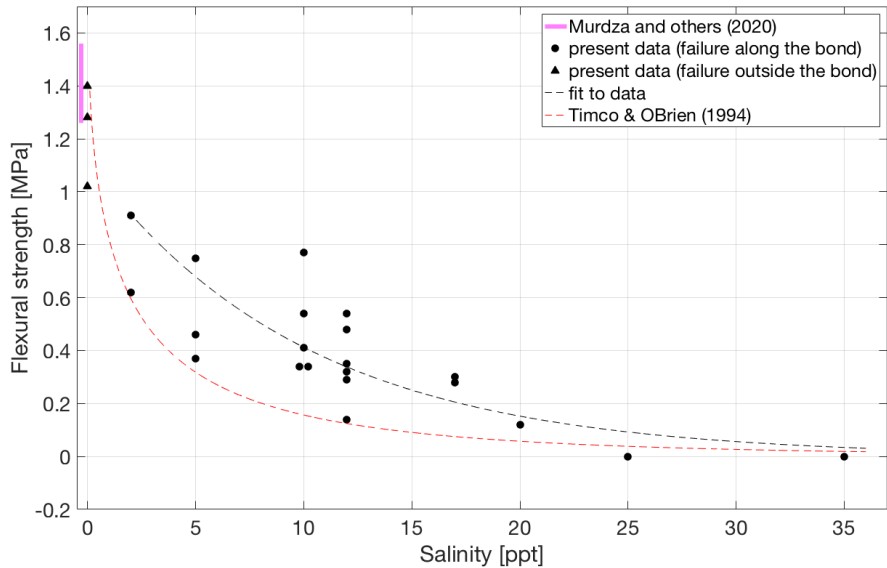


**Figure 3. Flexural strength at -3 °C of bonded ice as a function of the salinity of the salt water from which the bond was**
**formed. The solid pink line indicates the flexural strength 1.42±0.16 MPa of parent freshwater ice at -3 °C (Murdza and**
**others, 2020). A red dashed line is taken from Timco and O'Brien (1994) for the ice at -3 °C. A black dashed line is a fit to**
**the present data.**



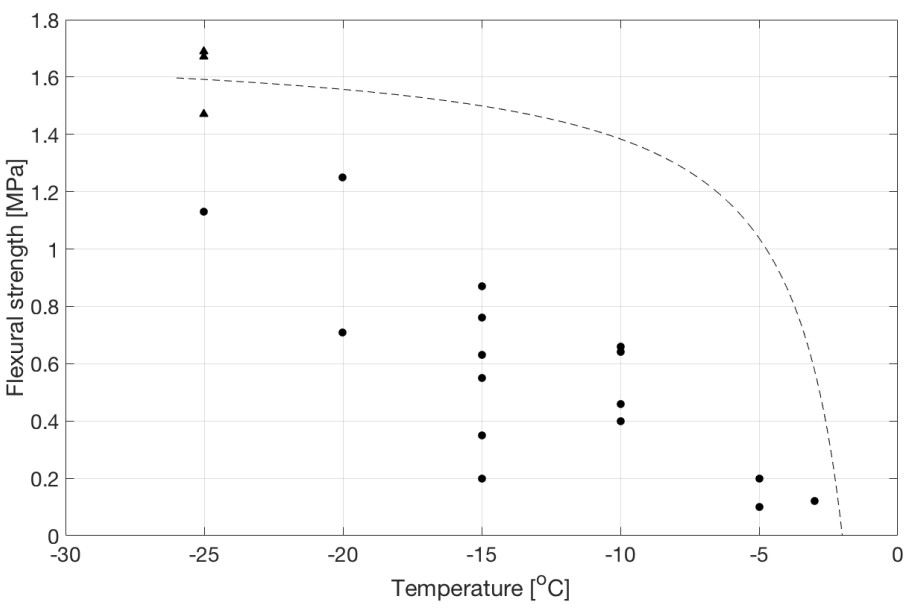


**Figure 4. Flexural strength of bonded ice as a function of temperature for bonds formed from water of salinity of 20 ppt.**
**Triangular-shaped points at -25 °C indicate that actual bond strength is greater than that of the parent material as the**
**failure occurred outside the bond. The dotted line is drawn according to the model in Appendix B.**

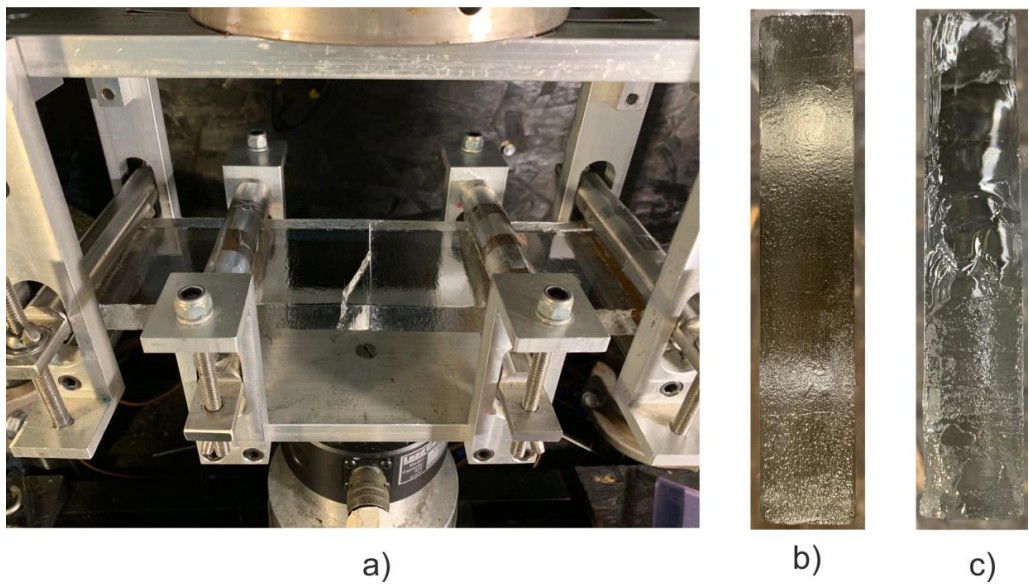


**Figure 5. Photographs of an ice sample #38 right after forced failure (a); the saline bond surface of 10 ppt after a crack**
**propagated fully through the bond, sample #19 (b) and partially through the parent material, sample #47 (c).**
