# Peer review of "The flexural strength of bonded ice"

_The Cryosphere, 2020_

## Referee Comment (RC1) · Anonymous Referee #1 · 21 Dec 2020

Review Comments

I find this paper to give an very useful review of the strength of the freeze-bond formed between two contacting ice surfaces. Also, and of great importance, the authors present important new data and observations regarding the freeze-bod strength of bonded ice.

The paper is easily read and its main conclusions clearly understood. I enjoyed reading the paper.

I recommend that this paper be accepted for publication after several minor adjustments (evident to me) are made. The following adjustments struck me:

Line 14: I had expected a section (in Results and Observations) on temperature-increase effects. Though temperature is mentioned, I did not see a discussion of how

temperature affects freeze-bond flexural strength. Perhaps, in line with the Abstract, such a section should be added.

Lines 31-40. The cohesion when using the Mohr-Coulomb model to describe bonding between ice blocks is often referred to as a "pseudo-cohesion", which to me indicates that the term "cohesion' is used as a convenience, with the actual nature of the "pseudo-convenience" left to be determined. "Cohesion" is a term understood in civil engineering and thus has its uses. I fully agree with the authors that "pseudo-cohesion" has been needed to be investigated, something that the authors have now largely accomplished. I do wonder, though whether the authors should mention "pseudo-cohesion" and the vagueness inherent in this term.

Line 56: How thick were the pucks?

Lines 210-211: Shouldn't semi-colons be used when listing? I.e., ...: (1) ...: (2) ...; and, (3)

Line 265: The authors sometimes use the second-person voice "one may..." when the remainder of the paper is in the third-person voice. I wonder is some consistency is needed.

Line 271: A space exists in front of the period ( .).

Line 297, Conclusions: Should the authors relate back to the objective of the paper (and the paper's Abstract) and be more definitive about bond-strength variation with temperature?

Again a very interesting and useful paper.

---

## Referee Comment (RC2) · Knut Høyland (Referee) · 31 Dec 2020

**Flexural strength of freeze-bonds**

**General comments**

This is a very interesting experimental study. It described what seems to be well conducted experiments and it is well written. However, to major things should be improved:

1. More in-depth explanation of experimental procedures.

2. The discussion should be improved. Below I suggest several interesting perspectives that could be used to analyse your nice results. But I understand if you do not have the time to include all these.

I am not sure if it is **major** or **minor** modifications, but I encourage you to improve the paper. I will be happy to review the revised manuscript.

**1. Introduction**

Be aware that also Szabo and Schneebeli (2007) did experimental studies of tensile strength of freeze-bonds. Their work was different than your, but it may be nice to read.

**2. Experimental procedure**

1. Lines 65-67. If I understand correctly you are making the freeze-bonds so that they simulate the strength of refrozen vertical cracks in the ice cover, and not the bond between layers of rafted ice? A simple sketch may help to clarify, it may also help to explain which natural physical mechanisms you are trying to mimic or address.

2. Lines 69-75. I don't understand how you treated the ice. How cold was the ice when it was sprayed with mist? Please explain carefully the procedure of storing and handling the ice. The reader may use this to try to understand the thermal regime of the ice. It was stored in -10°C, but the spraying of mist was done at +2°C. How long was the ice at +2°C before the mist was applied?

3. Lines 108-11. Which strain-rate does a loading rate of 0.1 mm/s in the outer-fiber (according to Linear-elastic first order beam theory I assume?) correspond with?

4. You indicate an interesting and important distinction between simplified beam theory and ordinary continuum mechanics. Would it be possible to give the analytical expressions for the continuum 3D stresses in a linear-elastic beam? You may use appendix to explain the derivation and possible correction for perfect plastic material.

   If you want you could bring this further by discussing different failure theories (max tensile stress, max difference between major and minor tensile stress, Coulomb-Mohr etc.) can explain the observations. This may be too much for this experimentally based paper, but I encourage you to study theoretical models and expand your work.

5. The statement about real major and minor stresses is interesting, but should be expanded (as suggested above) or deleted. You should at least let the reader know if all principle stresses are tensile. If we compare the continuum stress states in the same material in a tensile and beam test it may be possible to explain the experimental 1.7 factor. A very simple suggestion follows here. Let us assume Tresca failure criterion ($c$ as the material property) then the following applies at failure:

$$c = 1/2(\sigma_{major} - \sigma_{minor}) \tag{1}$$

For a tensile test there is only one non-zero principal stress:

$$c = 1/2(\sigma_{tensile} - 0) = 1/2\sigma_{tensile} \tag{2}$$

For a beam test there are both non-zero major and minor stresses and one gets:

$$c = 1/2(\sigma_{major} - \sigma_{minor}) = 1/2(\sigma_f - 1/3\sigma_f) = 1/3\sigma_f \tag{3}$$

Since the material ($c$) is the same in both tests this gives

$$\sigma_f/\sigma_{tensile} = 3/2 \tag{4}$$

I am not sure how good this suggestion is, but you may see if you think it makes sense. In reality the stress state in a tensile test is also affected by different stresses, but the match with the experimental data you refer to is veey good.

**3. Results and observations**

1. Lines 133. I suggest you remove Appendix 1 including all references. See below for why I think so.

2. Lines 133 continued. I am not sure if it is surprising that freeze-bonds formed under cold conditions become stronger than parent ice. It has been observed earlier by Høyland and Møllegaard (2014) and I believe also Shafrova and Høyland (2008). If the grains in the freeze-bond are smaller it may help to understand this phenomenon.

3. Lines 148-149. This is not an observation, so I suggest you move it to **Discussion**

**4. Discussion**

1. Lines 185-193. The observation around the 1.7 factor for $\sigma_f$ vs. $\sigma_t$ is interesting. It would be great if you cold argue a bit, at least suggest possible mechanisms. Here you could explain that there are several reasons for flexural strength to be different than tensile strength. Even if we forget the discussion above on the existence of a minor principle stress, the tensiel and flexural strength are equal **only** if linear-elastic first-order beam theory (LEFB) applies, and in reality it rarely does. We may distinguish between two things:

   • Three-dimensional continuum (real?) stresses are different than LEFB stresses. Earlier in the paper you hint about this, and if you could argue that LEFB theory systematically predicts higher strength it would be excellent. Anyway, please mention this possibility.

- If the vertical stress distribution is non-linear it will also give different strengths compared to LEFB. The flexural strength can be both higher and lower, see Ervik et al. (2014) for a discussion on this. This probably does not apply to your experiments as the beam were isothermal, but it applies since you compare with Timco and O'Brien (1994).

2. Lines 195-206. It is good that you summarize earlier literature, you could add Høyland and Møllegaard (2014), Marchenko and Chenot (2009), Szabo and Schneebeli (2007) and Bueide and Høyland (2015) to complete the list. The fact that you used freshwater ice and that your experiments in many ways were more carefully conducted when it comes to ice quality (same ice all the time) and the testing rig (compared to the ones I was involved in), it is not surprising that you find high values of strength. However, a short discussion around the word *strength* would be good. In the (rare) case of one-parameter models such as von-Mises the word strength may be appropriate, but even then this material parameter is generally not equal to the highest force with dimension of stress (such as equation 1 in this paper). In the literature the words strength sometimes means the highest force with unit of stress obtained in the experiments, and sometimes it means some kind of material parameter. This is very important in the shear experiments where the average shear force often do not represent the strength, see Bueide and Høyland (2015) for a discussion here.

3. Lines 221-231. This is not the first study that reports higher FB strength than parent material strength. There are several explanations of this, but in your case it could be that the FB would be more granular than the columnar ice and that the *strength* of tearing apart the columnar ice with larger grains would be less than the smaller more granular texture of the FB?

4. FB and presence of water. It seems that the presence of water is essential in the formation of freeze-bond. Szabo and Schneebeli (2007) tested tensile strength of fresh-water freeze-bonds formed in air, and found stronger bonds for warmer ice. You add only a little bit of water, probably enough to create bonds, but not enough to heat the ice and this should produce some kind of maximal strength.

**5. Appendix 1**

I don't understand this appendix and I suspect that it is wrong. It starts with the Stefan condition and then claims to integrate it without explain neither which domain the integration yields nor the boundary conditions. I suggest you skip this appendix. Alternatively you can expand it and explain properly, but I suspect you will end of with another independent article. A major challenge with applying continuum thermodynamics is the quantification of the thickness of the bond.

**References**

Bueide, I. M., Høyland, K. V., 2015. Confined comprssion tests on saline ice and freeze-bonds. In: Proc. of the 23th Int. Conf. on Port and Ocean Eng. under Arctic Conditions (POAC), NTNU, Trondheim, Norway. paper #186.

Ervik, A., Høyland, K. V., Marchenko, A., Karulina, M., Karulin, E., 2014. In-situ experimental

investigations of the vertical stress distribution in sea ice covers; a comparison of tensile and flexural strength. In: Proc. of the 22 Int. Symp. on Ice (IAHR), Singapore. Paper #1113.

Høyland, K. V., Møllegaard, A., 2014. Mechanical behaviour of laboratory made freeze-bonds as a function of submersion time, initial ice temperature and sample size. In: Proc. of the 22 Int. Symp. on Ice (IAHR), Singapore. Paper #1202.

Marchenko, A., Chenot, C., 2009. Relegation of ice blocks in the water and on the air. In: Proc. of the 20th Int. Conf. on Port and Ocean Eng. under Arctic Conditions (POAC), Luleå, Sweden. Vol. POAC09-67.

Shafrova, S., Høyland, K. V., 2008. Morphology and 2D spatial strength distribution in two Arctic first-year sea ice ridges. Cold Regions Science and Technology (51), 38-55.

Szabo, D., Schneebeli, M., 2007. Subsecond sintering of ice. Applied Physics letter (90:151916), 3 p.

Timco, G. W., O'Brien, S., 1994. Flexural strength equations for sea ice. Cold Regions Science and Technology (22), 285-298.

---

## Author Comment (AC1) · 15 Feb 2021

**Responses to comments by reviewers of manuscript tc-2020-301-RC1 "The flexural strength of bonded ice"**

We sincerely thank anonymous referee for valuable comments/suggestions on our work. The comments are constructive and insightful. We have modified our manuscript according to them. Please, see all the responses in red.

**Comments:**

I find this paper to give an very useful review of the strength of the freeze-bond formed between two contacting ice surfaces. Also, and of great importance, the authors present important new data and observations regarding the freeze-bod strength of bonded ice.

The paper is easily read and its main conclusions clearly understood. I enjoyed reading the paper.

I recommend that this paper be accepted for publication after several minor adjustments (evident to me) are made. The following adjustments struck me:

Line 14: I had expected a section (in Results and Observations) on temperature increase effects. Though temperature is mentioned, I did not see a discussion of how temperature affects freeze-bond flexural strength. Perhaps, in line with the Abstract, such a section should be added.

Re: Thank you for this comment. We organized the structure of Results & Observations for bonded ice in such a way that we divide it only for two parts, i.e. Freshwater bond subsection and Saline bond subsection. Within each subsection we present results on the effect of temperature, salinity, time, etc. We do not think that it is reasonable to divided each subsection into a few more and to devote each of them to temperature, salinity and time observations separately. We do, however, present the temperature effect on both the freshwater and saline bond strength in Results and observation (for example, lines 172-181). We also provide tables and plots with data that provide explicitly the temperature dependence. The temperature effect on the flexural strength is mentioned in the Discussion in lines 280-286 and we also provide Appendix C where we derive an analytical equation for the strength vs temperature dependency. Therefore, we think that we covered the effect of temperature on the freeze-bond.

Lines 31-40. The cohesion when using the Mohr-Coulomb model to describe bonding between ice blocks is often referred to as a "pseudo-cohesion", which to me indicates that the term "cohesion' is used as a convenience, with the actual nature of the "pseudo-convenience" left to be determined. "Cohesion" is a term understood in civil engineering and thus has its uses. I fully agree with the authors that "pseudo-cohesion" has been needed to be investigated, something that the authors have now largely accomplished. I do wonder, though whether the authors should mention "pseudocohesion" and the vagueness inherent in this term.

Re: Thank you for this suggestion. The term "cohesion" is also commonly used the ice engineering literature related to freeze-bonds and partly consolidated ice rubble. Therefore, we decided to keep the term "cohesion" in order to be consistent with the literature.

Line 56: How thick were the pucks?

Re: We added to the sentence that the thickness of the ice puck is ~25 cm.

Lines 210-211: Shouldn't semi-colons be used when listing? I.e., . . .: (1) . . .: (2) . . .; and, (3)

Re: Thank you for this comment. We have corrected this.

Line 265: The authors sometimes use the second-person voice "one may..." when the remainder of the paper is in the third-person voice. I wonder is some consistency is needed.

Re: We have replaced "As one may have expected" with "As expected" in line 280.

Line 271: A space exists in front of the period ( .).

Re: Thank you for pointing this out. We deleted the period.

Line 297, Conclusions: Should the authors relate back to the objective of the paper (and the paper's Abstract) and be more definitive about bond-strength variation with temperature?

Re: Thank you for this comment. Conclusion 3 actually states how the bond strength changes with temperature: "An increase in bond salinity and **in freezing temperature** leads to a **decrease in bond strength**". We tried to keep conclusion concise, although such that includes all main observations. Therefore, given the structure of the conclusions we used, we do not want to include more information on bond-strength variation with temperature.

Again a very interesting and useful paper.

---

## Author Comment (AC2) · 15 Feb 2021

**Responses to comments by reviewers of manuscript tc-2020-301-RC2 "The flexural strength of bonded ice"**

We sincerely thank Prof. Knut Høyland for valuable comments/suggestions on our work. The comments are constructive and insightful. We have modified our manuscript according to them. Please, see all the responses in red.

**Comments:**

**General comments**

This is a very interesting experimental study. It described what seems to be well conducted experiments and it is well written. However, to major things should be improved:

1. More in-depth explanation of experimental procedures.

2. The discussion should be improved. Below I suggest several interesting perspectives that could be used to analyse your nice results. But I understand if you do not have the time to include all these.

I am not sure if it is **major** or **minor** modifications, but I encourage you to improve the paper. I will be happy to review the revised manuscript.

**1. Introduction**

Be aware that also Szabo and Schneebeli (2007) did experimental studies of tensile strength of freeze-bonds. Their work was different than your, but it may be nice to read.

Re: Thank you for pointing this out. We added this reference to our paper as this was also suggested by the reviewer's comments further below (comments 2 and 4 in Discussion section).

The last sentence of the first paragraph in Introduction now reads: "Szabo and Schneebeli (2007) performed tensile tests on sintered ice grains of scale ~$10^{-3}$ m, but to our knowledge, no data on freeze-bond strength under tensile loading **at time and length scales relevant to geophysical or ice engineering** problems have been published". It is true that Szabo and Schneebeli (2007) also conducted bonding experiments and investigated tensile strength as a function of time and temperature. However, the work by Szabo and Schneebeli (2007) is very small-scale work on sintering, and as such, quite different from our work. In addition, the contact times are on the scale of 10-1000 ms. S&S (2007) explain the time dependence of their "sintering force" on creep-depended increase in contact area and they observe opposite effect of temperature than what we see (this is also pointed out by the reviewer further below in comment 4 in Discussion section). S&S appear, or claim, to discuss the scale of an ice grain – our study is on a scale of, at least, tens or hundreds of grains, which also then allows reasoning for the bond strength being higher than the strength of the parent material (grain size within bond smaller than that of parent material).

**2. Experimental procedure**

1. Lines 65-67. If I understand correctly you are making the freeze-bonds so that they simulate the strength of refrozen vertical cracks in the ice cover, and not the bond between layers of rafted ice? A simple sketch may help to clarify, it may also help to explain which

natural physical mechanisms you are trying to mimic or address.

Re: Thank you for this comment. We agree with the reviewer's suggestion to clarify whether we simulated refrozen vertical cracks or the bond between layers of rafted ice and modified the sentence on lines 26-28 in the following way: "It is relevant to ask if the freeze bonds forming into vertical cracks within a broken and refrozen ice cover form the weakest link at which wave-induced cracks initiate and propagate". Further, in our opinion the bond location can be clearly seen in Figure 1 and, therefore, we did not add an additional sketch.

2. Lines 69-75. I don't understand how you treated the ice. How cold was the ice when it was sprayed with mist? Please explain carefully the procedure of storing and handling the ice. The reader may use this to try to understand the thermal regime of the ice. It was stored in -10°C, but the spraying of mist was done at +2°C. How long was the ice at +2°C before the mist was applied?

Re: Thank you for this comment. To make the paragraph clearer, we added two sentences. The first sentence is at the beginning of the paragraph as follows: "The two parts of the specimen were then placed in a cold room with a temperature of +2°C for a few minutes". The second sentence we added at the end of this paragraph: "Afterwards, the freeze-bonding rig was moved to another cold room with a desired test temperature". Additionally, line 69 describes that before initiation of a freeze-bond formation "specimens were allowed to equilibrate to the test temperature", which ranged from -3C to -25C, depending on the experiment.

3. Lines 108-11. Which strain-rate does a loading rate of 0.1 mm/s in the outer-fiber (according to Linear-elastic first order beam theory I assume?) correspond with?

Re: This information is provided in lines 112-113: "The experiments were performed at an outer-fiber center-point displacement rate of 0.1 mm s-1 (or outer-fiber strain rate of about 1.4 x 10-4 s-1).".

4. You indicate an interesting and important distinction between simplified beam theory and ordinary continuum mechanics. Would it be possible to give the analytical expressions for the continuum 3D stresses in a linear-elastic beam? You may use appendix to explain the derivation and possible correction for perfect plastic material.

If you want you could bring this further by discussing different failure theories (max tensile stress, max difference between major and minor tensile stress, Coulomb-Mohr etc.) can explain the observations. This may be too much for this experimentally based paper, but I encourage you to study theoretical models and expand your work.

Re: Thank you for this comment. We provide in the manuscript equation 1 for the major outer-fiber stress in the ice plate bent under 4-point bending under linear-elastic beam theory assumptions. This is a well-known formula and we believe it does not require derivation. We added the derivation of the relationship between major and minor stresses in the ice plate (given in Appendix A) assuming both elasticity and plasticity theories (the author also mentioned this in his next comment). Regarding the second part of the reviewer's comment about discussing different failure theories: Scope of this manuscript was to document and present experimental results on flexural strength of freeze bonds and, thus, we leave the study on failure criterions for future work as such investigation would likely require experimentation under a number of different stress states.

5. The statement about real major and minor stresses is interesting, but should be expanded (as suggested above) or deleted. You should at least let the reader know if all principle stresses are tensile. If we compare the continuum stress states in the same material in a tensile and beam test it may be possible to explain the experimental 1.7 factor. A very simple suggestion follows here. Let us assume Tresca failure criterion ($c$ as the material property) then the following applies at failure:

$$c = 1/2(\sigma_{major} - \sigma_{minor}) \tag{1}$$

For a tensile test there is only one non-zero principal stress:

$$c = 1/2(\sigma_{tensile} - 0) = 1/2\sigma_{tensile} \tag{2}$$

For a beam test there are both non-zero major and minor stresses and one gets:

$$c = 1/2(\sigma_{major} - \sigma_{minor}) = 1/2(\sigma_f - 1/3\sigma_f) = 1/3\sigma_f \tag{3}$$

Since the material ($c$) is the same in both tests this gives

$$\sigma_f / \sigma_{tensile} = 3/2 \tag{4}$$

I am not sure how good this suggestion is, but you may see if you think it makes sense. In reality the stress state in a tensile test is also affected by different stresses, but the match with the experimental data you refer to is veey good.

Re: Thank you for this comment. We added Appendix A where we show derivations for the relationship between minor and major stress. In the text we mention that both principal stresses are tensile ("a biaxial state of tension developed in the ice").

**3. Results and observations**

1. Lines 133. I suggest you remove Appendix 1 including all references. See below for why I think so.

Re: Thank you for this comment. We agree with the reviewer in that the derivations provided in Appendix A (now Appendix B) are not very detailed. To improve this appendix according to the reviewer's comment, we indicated intervals over which we integrated equation A1. We think that the appendix is worth keeping, since its goal is to give a rough estimate of the time for the freeze-bond formation (by the order of magnitude) and compare it with experimental observation rather than to develop a sophisticated precise theoretical model, which may take significant extra efforts and what seems to be suggested by the reviewer. We think, however, that developing detailed model for bond formations is a great idea and could be a topic of another manuscript.

2. Lines 133 continued. I am not sure if it is surprising that freeze-bonds formed under cold conditions become stronger than parent ice. It has been observed earlier by Høyland and Møllegaard (2014) and I believe also Shafrova and Høyland (2008). If the grains in the freeze-bond are smaller it may help to understand this phenomenon.

Re: Thank you for this comment. We found that the results in Høyland and Møllegaard (2014) provide an indication that the freeze-bonds could become stronger than the parent material; however, we did not find similar results from Shafrova and Høyland (2008). The results in Høyland and Møllegaard (2014) relate to experiments, where freeze bonds sheared under compression on an unconfined bonded sample, and in some cases, instead of shearing the samples

split. Thus, there was a change in the failure mode of the sample. It is anyhow challenging to justify, whether the shear strength of the bond in this case actually exceeded the strength of the parent material or just got high enough to to allow axial splitting instead of shear-like failure of the sample. We therefore think this outcome is different from ours, as we directly show that the tensile strength of the bond is higher than that of the parent material and, therefore, we think that our results are novel. We added a reference to Høyland and Møllegaard (2014) to the Discussion section: "**Aligning with our observations, the results by Høyland and Møllegaard (2014) provide an indication of the shear strength of freeze bonds reaching strengths comparable to that of the parent material. In their uniaxial compression tests on bonded cylindrical samples having an inclined freeze-bond, the failure changed from shearing (along the plane of the bond) to axial splitting of the sample in some of the cases.**"

**3.** Lines 148-149. This is not an observation, so I suggest you move it to **Discussion**

Re: This was moved to Discussion.

**4. Discussion**

1. Lines 185-193. The observation around the 1.7 factor for $\sigma_f$ vs. $\sigma_t$ is interesting. It would be great if you cold argue a bit, at least suggest possible mechanisms. Here you could explain that there are several reasons for flexural strength to be different than tensile strength. Even if we forget the discussion above on the existence of a minor principle stress, the tensiel and flexural strength are equal **only** if linear-elastic first-order beam theory (LEFB) applies, and in reality it rarely does. We may distinguish between two things:

   - Three-dimensional continuum (real?) stresses are different than LEFB stresses. Earlier in the paper you hint about this, and if you could argue that LEFB theory systematically predicts higher strength it would be excellent. Anyway, please mention this possibility.

   - If the vertical stress distribution is non-linear it will also give different strengths compared to LEFB. The flexural strength can be both higher and lower, see Ervik et al. (2014) for a discussion on this. This probably does not apply to your experiments as the beam were isothermal, but it applies since you compare with Timco and O'Brien (1994).

Re: The factor of 1.7 between tensile and flexural strength is obtained from statistical analysis, specifically, Weibull distribution. This is a fairly well-known result and, therefore, we provide a reference to a textbook by Ashby and Jones (2012) where this relationship is discussed. We also added a sentence to the manuscript where we explain in more detail why the flexural strength is greater than the tensile strength: "The reason is that in bending only a thin layer close to one surface of the sample(and thus a relatively small volume) carries the peak tensile stress and it is less likely that this volume contains larger flaw, while in tension the entire sample carries the tensile stress and it is more likely that it will contain larger flaws".

Regarding the second part of the comment and reference to Ervik et al. (2014), we agree that  in the field bending experiments that are presented in this paper, vertical stress distribution may be non-linear. However, we would like to point out a few differences between the experiments described in the present paper and those in Ervik et al. (2014). In the field experiments described in  Ervik et al. (2014) there was a temperature gradient through the ice thickness. We assume that

this may lead to the variation of Young's modulus across the beam thickness and as a result may lead to the non-linear vertical stress distribution. In our experiments, however, the ice specimens were isothermal, i.e. at constant temperature and therefore, we believe that the Young's modulus was constant through thickness. In addition, in our experiments both parts of the specimen were always fresh and only the bond salinity was varied, while the experiments in Ervik et al. (2014) were conducted on sea ice.

Given these differences, and also the fact that the goal of this work was to investigate the strength of bonded ice rather than to run extensive analysis on the proper usage and possible corrections of Equation 1 under different conditions, we believe that we can use the Equation 1 as it is commonly used in the literature assuming linear elastic theory.

2. Lines 195-206. It is good that you summarize earlier literature, you could add Høyland and Møllegaard (2014), Marchenko and Chenot (2009), Szabo and Schneebeli (2007) and Bueide and Høyland (2015) to complete the list. The fact that you used freshwater ice and that your experiments in many ways were more carefully conducted when it comes to ice quality (same ice all the time) and the testing rig (compared to the ones I was involved in), it is not surprising that you find high values of strength. However, a short discussion around the word *strength* would be good. In the (rare) case of one-parameter models such as von-Mises the word strength may be appropriate, but even then this material parameter is generally not equal to the highest force with dimension of stress (such as equation 1 in this paper). In the literature the words strength sometimes means the highest force with unit of stress obtained in the experiments, and sometimes it means some kind of material parameter. This is very important in the shear experiments where the average shear force often do not represent the strength, see Bueide and Høyland (2015) for a discussion here.

Re: Thank you for this comment. We added a sentence (lines 118-119) where we explicitly state what we mean by the word "strength": "The flexural strength that we refer to throughout this paper is the maximum major outer-fiber stress that the ice plate can withstand before breaking".

We also added suggested references to Discussion.

3. Lines 221-231. This is not the first study that reports higher FB strength than parent material strength. There are several explanations of this, but in your case it could be that the FB would be more granular than the columnar ice and that the *strength* of tearing apart the columnar ice with larger grains would be less than the smaller more granular texture of the FB?

Re: Thank you for this comment. We added a sentence to our manuscript where we indicate that this effect was earlier observed by Høyland and Møllegaard (2014): "**Aligning with our observations, the results by Høyland and Møllegaard (2014) provide an indication of the shear strength of freeze bonds reaching strengths comparable to that of the parent material. In their uniaxial compression tests on bonded cylindrical samples having an inclined freeze-bond, the failure changed from shearing (along the plane of the bond) to axial splitting of the sample in some of the cases.**" Regarding the reasons for the bond strength to be higher than the strength of the parent material, the explanation is given in lines 267-273 and it is the same as suggested in this comment.

4. FB and presence of water. It seems that the presence of water is essential in the formation of freeze-bond. Szabo and Schneebeli (2007) tested tensile strength of fresh-water freeze-bonds formed in air, and found stronger bonds for warmer ice. You add only a little bit of water, probably enough to create bonds, but not enough to heat the ice and this should produce some kind of maximal strength.

Re: Thank you very much for pointing this out. It is correct that Szabo and Schneebeli (2007) conducted bonding experiments in air without adding any water. The authors suggested that the increase in strength is most likely due to the presence of the liquidlike layer on the surface of ice, and, therefore, ice at warmer temperatures resulted in higher strength values since at higher temperatures the liquid-like layer is thicker. However, the experiments were conducted over time periods of 10-1000 ms, which is substantially different from our experiments where time periods were in the order of hours. Therefore, we think that the mechanisms responsible for increase in strength in our work and in Szabo and Schneebeli (2007) are different, as evidenced by the fact that the increase in bond strength in our experiments is greater and faster at lower temperatures compared with warmer temperatures, which is different from Szabo and Schneebeli (2007). However, by saying that we do not want to exclude sintering mechanism from our experiments which is responsible for strengthening and described in Szabo and Schneebeli (2007) as it may also take place in our experiments, although to a lower degree.

**5. Appendix 1**

I don't understand this appendix and I suspect that it is wrong. It starts with the Stefan condition and then claims to integrate it without explain neither which domain the integration yields nor the boundary conditions. I suggest you skip this appendix. Alternatively you can expand it and explain properly, but I suspect you will end of with another independent article. A major challenge with applying continuum thermodynamics is the quantification of the thickness of the bond.

Re: We already responded to this comment above and have slightly modified the appendix. Goal of the appendix is to give rough estimate, the order of magnitude, for the time for the freeze-bond formation. It is true that the thickness of the freeze-bond is difficult to estimate, or even measure.

**References**

Bueide, I. M., Høyland, K. V., 2015. Confined comprssion tests on saline ice and freeze-bonds. In: Proc. of the 23th Int. Conf. on Port and Ocean Eng. under Arctic Conditions (POAC), NTNU, Trondheim, Norway. paper #186.

Ervik, A., Høyland, K. V., Marchenko, A., Karulina, M., Karulin, E., 2014. In-situ experimental investigations of the vertical stress distribution in sea ice covers; a comparison of tensile and flexural strength. In: Proc. of the 22 Int. Symp. on Ice (IAHR), Singapore. Paper #1113.

Høyland, K. V., Møllegaard, A., 2014. Mechanical behaviour of laboratory made freeze-bonds as a function of submersion time, initial ice temperature and sample size. In: Proc. of the 22 Int. Symp. on Ice (IAHR), Singapore. Paper #1202.

Marchenko, A., Chenot, C., 2009. Relegation of ice blocks in the water and on the air. In: Proc. of the 20th Int. Conf. on Port and Ocean Eng. under Arctic Conditions (POAC), Luleå, Sweden. Vol. POAC09-67.

Shafrova, S., Høyland, K. V., 2008. Morphology and 2D spatial strength distribution in two Arctic first-year sea ice ridges. Cold Regions Science and Technology (51), 38-55.

Szabo, D., Schneebeli, M., 2007. Subsecond sintering of ice. Applied Physics let- ter (90:151916), 3 p.

Timco, G. W., O'Brien, S., 1994. Flexural strength equations for sea ice. Cold Regions Science and Technology (22), 285-298.

---

## Referee Report (RR1)

**Flexural strength of freeze-bonds R1**

**General comments**

This is a very interesting experimental study. It described what seems to be well conducted experiments and it is well written. However, there are still important things that should be improved as given below.

**Vital things to improve**

1. My encouragement about expanding the discussion to include the relationship between linear-elastic brittle beam theory (equation 1 in the paper) and other theories such as 2 or 3-D stress-failure criteria has not been done. It is not essential for the paper as it is mostly an experimental study, but would make the paper more interesting and useful. In appendix A you have included Hookes law and a plastic development law (flow rule). None of these are (or should be!) compared with Equation 1. If you want to do this you need to include equation for yield (or failure) criterion, not pre-failure (elastic) or post-failure (flow rule). To get the paper accepted you should either remove the appendix and the text in lines 122 - 124. Or you should include one, or more equations for yield (or failure) and compare these with the beam theory. **I would like to underline that it is OK if you don't want to spend time on this and leave it out of the paper**.

2. The words strength. It is not correct that you use flexural strength about the peak stress in the outer fiber. You use the word flexural strength about the value you get from equation 1. Further you argue that this is probably similar to the extreme stress in the outer fiber. There is an important difference here (se point further down) that I think you should use the opportunity to explain.

3. Appendix B. I can only recommend that this paper is published if you take way Appendix B and all the references to it. In the derivation you have assumed that Fourier's number characterizes the process without any discussion or justification. You could have assumed a linear relationship between time and length and the practical predictions of the time would equally well fit your data. In other words, the derivation does not help building credibility in your results. It is not so that I don't believe in your results, on the contrary I think they are good and solid. But, you should avoid arguing without substance. I am sorry to be so categoric, but it is important to compare with theory in a rational way. If one really wants one can make most theories fit a set of experiments.

**Also please improve**

Tensile and flexural failure. It is good that you enhanced the discussion on this. But, it would be great if you could explain the very simple fact that these should only be equal if the material is linear-elastic and brittle (takes one sentence and a reference). Then you could explain that

in your case the linear-elastic-brittle assumption is probably fine and the explanation have to be elsewhere. If you read in the literature there are many ice-paper where linear-elastic-brittle assumption is used on analysis of field data without discussion. In this paper you have a chance the help to enlighten thew ice mechanical community on this and I hope you use this opportunity.

---

## Author Response (AR2)

**Responses to comments by reviewers of manuscript tc-2020-301 "The flexural strength of bonded ice"**

We thank anonymous referees for additional comments on our work. We have modified our manuscript according to them. Please, see all the responses in red.

**Comments from Referee # 2**

Comment 1. My encouragement about expanding the discussion to include the relationship between linear-elastic brittle beam theory (equation 1 in the paper) and other theories such as 2 or 3-D stress-failure criteria has not been done. It is not essential for the paper as it is mostly an experimental study, but would make the paper more interesting and useful. In appendix A you have included Hookes law and a plastic development law (flow rule). None of these are (or should be!) compared with Equation 1. If you want to do this you need to include equation for yield (or failure) criterion, not pre-failure (elastic) or postfailure (flow rule). To get the paper accepted you should either remove the appendix and the text in lines 122 - 124. Or you should include one, or more equations for yield (or failure) and compare these with the beam theory. **I would like to underline that it is OK if you don't want to spend time on this and leave it out of the paper**.

We removed Appendix A and text in lines 122-124 as recommended. Since our work is an experimental study we decided not to focus on the development of precise theoretical models.

Comment 2. The words strength. It is not correct that you use flexural strength about the peak stress in the outer fiber. You use the word flexural strength about the value you get from equation 1. Further you argue that this is probably similar to the extreme stress in the outer fiber. There is an important difference here (se point further down) that I think you should use the opportunity to explain.

Tensile and flexural failure. It is good that you enhanced the discussion on this. But, it would be great if you could explain the very simple fact that these should only be equal if the material is linear-elastic and brittle (takes one sentence and a reference). Then you could explain that in your case the linear-elastic-brittle assumption is probably fine and the explanation have to be elsewhere. If you read in the literature there are many ice-paper where linear-elastic-brittle assumption is used on analysis of field data without discussion. In this paper you have a chance the help to enlighten thew ice mechanical community on this and I hope you use this opportunity.

We added clarification that we assume linear-elastic and brittle material when compare tensile and flexural strength (lines 119 and 194). We also provided the reference to the work by Ashby and Jones (2012), as recommended, where the authors compare flexural and tensile strength and indicate that the material is brittle linear-elastic.

Comment 3. Appendix B. I can only recommend that this paper is published if you take way Appendix B and all the references to it. In the derivation you have assumed that Fourier's number characterizes the process without any discussion or justification. You could have assumed a linear relationship between time and length and the practical predictions of the time would equally well fit your data. In other words, the derivation does not help building credibility in your results. It is not so that I don't believe in your results, on the contrary I think they are good and solid. But, you should avoid arguing without substance. I am sorry to be so categoric, but it is important to compare with theory in a rational way. If one really wants one can make most theories fit a set of experiments.

We removed Appendix B and all the references to it as recommended.